# LEVERAGING OBJECT DETECTION FOR DIVERSE AND ACCURATE LONG-HORIZON EVENTS FORECASTING

## ABSTRACT

Long-horizon event forecasting is critical across various domains, including retail, finance, healthcare, and social networks. Traditional methods, such as Marked Temporal Point Processes (MTPP), often rely on autoregressive models to predict multiple future events. However, these models frequently suffer from issues like converging to constant or repetitive outputs, which limits their effectiveness and general applicability. To address these challenges, we introduce DeTPP (Detection-based Temporal Point Processes), a novel approach inspired by a matching-based loss function from object detection. DeTPP employs a unique matching-based loss function that selectively prioritizes reliably predictable events, improving the accuracy and diversity of predictions during inference. Our method establishes a new state-of-the-art in long-horizon event forecasting, achieving up to a 77% relative improvement over existing MTPP and next-K methods. Furthermore, DeTPP enhances next-event prediction accuracy by up to 2.7% on a large transactions dataset and demonstrates high computational efficiency during inference. The implementation of DeTPP is publicly available on GitHub[1].

## 1 INTRODUCTION

Data from various domains, including internet activity, e-commerce transactions, retail operations, and clinical visits, is often recorded as timestamps and associated information. When ordered by their timestamps, these data points form event sequences, and it is crucial to develop methods capable of handling these complex data streams. Event sequences differ fundamentally from other data types. Unlike tabular data (Wang & Sun, 2022), events include timestamps and have an inherent order. In contrast to time series data (Lim & Zohren, 2021), event sequences are characterized by irregular time intervals and additional attributes. These distinctions necessitate the development of specialized models.

The primary task in the domain of event sequences is predicting future event types and their occurrence times. Indeed, the ability to accurately forecast sequential events is vital for applications such as stock price prediction, personalized recommendation systems, and early disease detection. The simplified domain, represented as pairs of event types and times, is typically called Marked Temporal Point Processes (MTPP) (Rizoiu et al., 2017). Additionally, structured modeling of dependencies between different data fields (McDermott et al., 2024) can be considered an extension of MTPP.

Practical applications often require predicting multiple future events within a specified time horizon. This task presents unique challenges that differ from traditional next-event prediction. The conventional approach typically relies on autoregressive models, which predict the next event step by step. While these models are effective for immediate next-event forecasting, their performance tends to deteriorate as the prediction horizon extends (Karpukhin et al., 2024).

This study identifies and addresses significant limitations of the autoregressive approach in the context of long-horizon prediction. In particular, autoregressive models accumulate errors over time, leading to constant or repetitive outputs. Additionally, their inference parallelism is limited due to dependencies on the latest predictions. To address these limitations, we propose DeTPP, a novel

---

[1] https://github.com/anonymous-9485560/mtpp-detection-iclr25

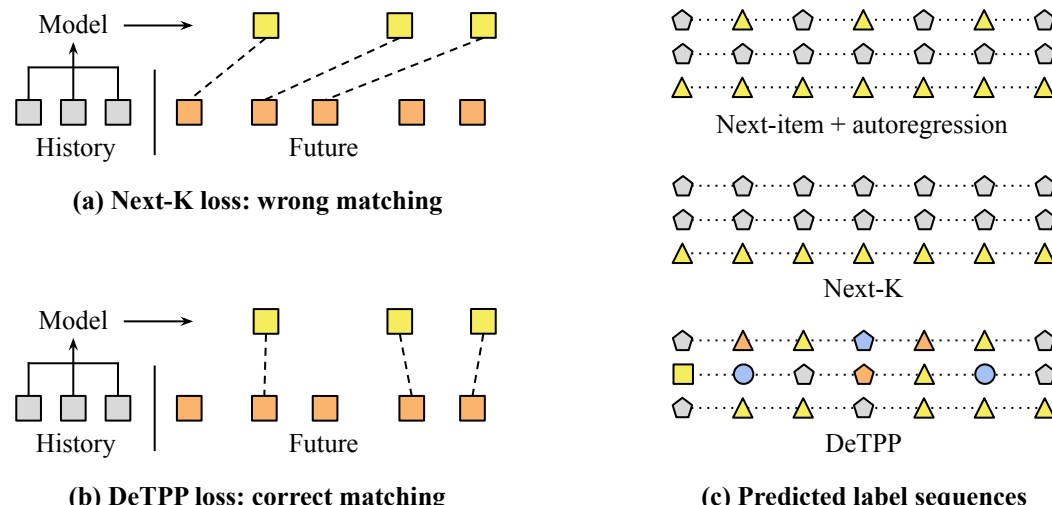

Figure 1: (a) A typical next-item or next-K loss compares events at corresponding positions, often resulting in incorrect matches. (b) The proposed matching loss calculates the loss function between the closest events, leading to a more robust and balanced error measure. (c) The proposed DeTPP method enhances prediction diversity. The illustration demonstrates 3 predicted sequences from the Amazon dataset for the autoregressive IFTPP method, its next-K extension, and the proposed DeTPP method. Each label type is depicted using a distinct shape and color combination. The precise timestamps are omitted for simplicity.

approach inspired by matching-based loss function from object detection. DeTPP predicts multiple future events in parallel. The novel matching loss, illustrated in Fig. 1, omits events that are inherently hard to model and focuses on robust modeling of other events. As a result, DeTPP sets a new state-of-the-art in long-horizon prediction, outperforming both autoregressive and next-K approaches. We also introduce a simple extension that combines ideas from DeTPP with traditional methods, improving next-event prediction quality, particularly on the large Transactions dataset.

## 2 RELATED WORK

**Marked Temporal Point Processes.** MTPP is a stochastic process that consists of a sequence of time-event pairs $(t_1, l_1), (t_2, l_2), \ldots$, where $t_1 < t_2 < \ldots$ denote the times of events, and $l_i \in \{1, \ldots, L\}$ are the corresponding event type labels (Rizoiu et al., 2017). Traditional MTPP models primarily focus on predicting the next event in the sequence. A straightforward approach is to independently predict the time and type of the next event, while more sophisticated methods model the temporal dynamics of each event type separately. These models often rely on Temporal Point Processes, which describe the stochastic generation of event times.

**MTPP models.** Traditional MTPP models, such as Poisson and Hawkes processes (Rizoiu et al., 2017), rely on strong assumptions about the underlying generative processes. Recent advancements have shifted towards more flexible and expressive models that leverage neural architectures. These include classical Recurrent Neural Networks (RNNs) (Du et al., 2016; Xiao et al., 2017; Omi et al., 2019; Shchur et al., 2019), as well as more advanced architectures like transformers (Zhang et al., 2020; Zuo et al., 2020; Wang & Xiao, 2022; Yang et al., 2022). Additionally, continuous-time models such as Neural Hawkes Processes (Mei & Eisner, 2017), ODE-RNN (Rubanova et al., 2019), and their variants (Jia & Benson, 2019; De Brouwer et al., 2019; Kidger et al., 2020; Song et al., 2024; Kuleshov et al., 2024) have been developed to capture the dynamics of event sequences better. Moreover, some researchers have adapted generative models, including denoising diffusion and Generative Adversarial Networks (GANs), for use in TPPs (Lin et al., 2022).

**Next-K models.** Previous research has explored models that predict multiple future events simultaneously, known as next-K models (Karpukhin et al., 2024). These models are typically trained using pairwise losses that match predicted events to ground truth events at corresponding positions.

**Long-horizon models.** The problem of long-horizon prediction has been addressed explicitly by HYPRO (Xue et al., 2022), which introduces a technique for selecting the best candidate from a set of generated sequences. HYPRO functions as a meta-algorithm that can enhance the performance of nearly any sequence prediction model. However, HYPRO's approach requires multiple generation runs for each prediction, significantly reducing training and inference speed.

## 3 LIMITATIONS OF AUTOREGRESSIVE INFERENCE

Previous studies have identified several challenges associated with autoregressive models for long-horizon predictions (Karpukhin et al., 2024). These models often exhibit reduced prediction uncertainty over extended horizons, even though the task becomes increasingly difficult. As illustrated in Fig. 1.c, the predicted label sequences often have constant or repetitive outputs. This behavior likely stems from the model's reliance on its predictions as input for subsequent predictions, which can amplify errors and lead to repeated events. Notably, even the next-K approach, despite avoiding autoregressive dependencies, exhibits repetitive patterns. This may be attributed to each head in the next-K model predicting the entire distribution of labels, leading to a bias toward the most frequent classes during inference.

In contrast, our proposed DeTPP approach exhibits greater diversity in its predictions and, as we will demonstrate, achieves superior performance in long-horizon prediction tasks. The details of the proposed method are discussed in the following section.

## 4 EVENT DETECTION WITH DETPP

This section introduces DeTPP, a novel approach to MTPP modeling that incorporates the loss function, motivated by DeTR object detection approach (Carion et al., 2020). DeTPP begins by utilizing a backbone model to extract embeddings from historical data. In the next stage, DeTPP predicts K future event candidates within a specified time horizon H, where K is larger than the typical sequence length. During training, the model aligns its predictions with ground truth and computes pairwise losses, as illustrated in Fig. 2. At inference time, the model retains only candidates with high prediction scores. Below, we provide a detailed overview of the event prediction head, the sequence model, and the associated training and inference procedures.

### 4.1 PREDICTION HEAD

DeTPP captures the complexity of event sequences by modeling each component of an event using a probabilistic framework. Specifically, DeTPP predicts the probability of an event occurring, the distribution of event labels, and the distribution of the time shift relative to the last observed event.

As depicted in Fig. 2, the probability $\hat{o}$ of an event occurring is modeled using a neural network with a sigmoid activation function. A separate head with softmax activation (SM) models the distribution $\hat{p}(l)$ of event labels. For the time shift, we employ an approach similar to Mixture Density Networks (MDNs) (Bishop, 1994) and intensity-free TPP (Shchur et al., 2019), modeling the time shift as a Laplace distribution with a unit scale parameter:

$$\mathrm{P}(t) = \frac{1}{2} e^{-|t - \hat{t}|}, \tag{1}$$

where $\hat{t}$ is the predicted time shift. This formulation offers a probabilistic interpretation of the MAE loss function. By combining the predicted probabilities, we can estimate the likelihood of a future event given the output of the model:

$$\log \mathrm{P}(y) = \log \hat{o} + \log \hat{p}(l) - |t - \hat{t}| - \log 2. \tag{2}$$

where $y = (t, l)$ represents an event with timestamp $t$ and label $l$. The probability of a missed event (i.e., no event occurring) is given by:

$$\log \mathrm{P}(\emptyset) = \log(1 - \hat{o}) + C_\emptyset, \tag{3}$$

Figure 2: DeTPP predicts K future events. Each prediction includes presence probability $\hat{o}$, time $\hat{t}$, and labels distribution $\hat{p}(l)$. During training, a special matching loss aligns predictions with the ground truth sequence and evaluates its likelihood.

where $C_\emptyset$ is a constant independent of the model's output, representing the probability associated with a reserved "unknown" time and label values. To compute the loss function, we omit $C_\emptyset$ since it does not influence the gradient during training.

This proposed probabilistic framework offers a rigorous basis for evaluating the likelihood of ground truth event sequences.

### 4.2 HORIZON MATCHING LOSS

DeTPP is designed to predict K future events $\{\hat{y}_i\}_{i=1}^K$ within the horizon H, defined as the interval $(t, t + \mathrm{H})$, where $t$ is the timestamp of the last observed event. The set of ground truth events within this horizon is denoted by $\{y_i\}_{i=1}^T$, where T may vary. The model aligns the predicted sequence with the ground truth sequence by finding the matching that minimizes the following loss function:

$$\mathcal{L}(y, \hat{y}) = \min_{\sigma \in \mathcal{A}} \left[ \sum_{i=1}^T \mathcal{L}_{\mathrm{pair}}(y_i, \hat{y}_{\sigma(i)}) + \mathcal{L}_{\mathrm{BCE}}(\sigma, \hat{y}) \right], \tag{4}$$

where $\mathcal{A}$ is the set of all possible alignments between the ground truth and predicted sequences and $\sigma$ represents a specific alignment. The optimal matching is computed using the Hungarian algorithm.

The pairwise loss $\mathcal{L}_{\mathrm{pair}}$ is similar to the negative log-likelihood of the ground truth event $y_i$ given the predicted distribution $\hat{y}_{\sigma(i)}$. Specifically:

$$\mathcal{L}_{\mathrm{match}}(y_i, \hat{y}_{\sigma(i)}) = |t_i - \hat{t}_{\sigma(i)}| - \log \hat{p}_{\sigma(i)}(l_i), \tag{5}$$

where $y = (t, l)$ is a ground truth event, $\hat{t}$ is the predicted timestamp, and $\hat{p}(l)$ is the predicted probability of the correct label. The binary cross-entropy loss $\mathcal{L}_{\mathrm{BCE}}$ is used to train the model to predict the presence probability of events:

$$\mathcal{L}_{\mathrm{BCE}}(\sigma, \hat{y}) = -\sum_{i \in \sigma} \log \hat{o}_i - \sum_{i \notin \sigma} \log(1 - \hat{o}_i), \tag{6}$$

where $\hat{o}_i$ is the predicted probability that the $i$-th event is matched with some ground truth event.

By minimizing the overall loss $\mathcal{L}(y, \hat{y})$, DeTPP is trained to accurately predict the parameters of the ground truth sequence, adapting to sequences of varying lengths, up to a maximum of K events. Unlike object detection training objectives such as DeTR Carion et al. (2020), DeTPP employs the same loss function for both the matching process and model training. Specifically, DeTPP integrates the alignment loss $\mathcal{L}_{\mathrm{BCE}}$ into the matching cost. As demonstrated in our experiments, this approach enhances the model's performance, especially on small datasets.

To enable the method to accurately address both the next-event prediction task and long-horizon forecasting, we incorporate the IFTPP loss function into the first output head. The final training objective is defined as follows:

$$\mathcal{L}_{\mathrm{DeTPP}}(y, \hat{y}) = \mathcal{L}_{(}y, \hat{y}) + \lambda \left[ |t_1 - \hat{t}_1| - \log \hat{p}_1(l_1) \right]. \tag{7}$$

### 4.3 CONDITIONAL HEAD ARCHITECTURE

Implementing a separate feed-forward network for each prediction head results in many parameters and increases the risk of overfitting. A common strategy to reduce the parameter count is applying a transformer decoder network to a set of $K$ queries, where the decoder shares the same projection matrices for all input vectors, lowering the total number of weights. However, since our model generates the entire set of predictions from a single embedding vector, we can simplify this approach by omitting the cross-attention layer.

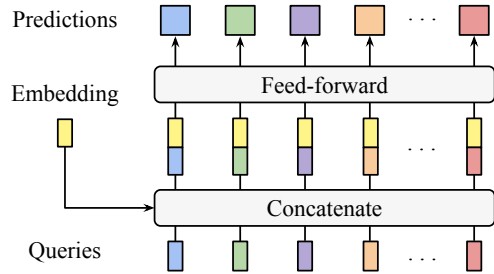

Figure 3: The DeTPP conditional prediction head.

In DeTPP, we implement a conditional feed-forward network, as shown in Fig. 3. The model takes two inputs: a query and a context vector. It then applies a feed-forward network to the concatenated input vectors. This way, the trainable query vector encodes the necessary information about each output head, allowing the same feed-forward network to generate all $K$ outputs. Using a single network instead of $K$ distinct ones significantly reduces the overall number of parameters, speeds up convergence, and improves prediction quality, as demonstrated in our ablation studies.

### 4.4 CALIBRATION AND INFERENCE

Inference with DeTPP involves two steps: filtering and sorting. First, predicted events are filtered based on their occurrence probabilities $\hat{o}_i$. The remaining events are then sorted according to their predicted timestamps, forming the final output sequence. However, in practice, an additional calibration step is necessary. Without calibration, the model tends to predict a small number of events due to a bias in the predicted occurrence probabilities $\hat{o}_i$ toward the matching frequency of each head, which is typically below the 0.5 threshold. Calibration aims to determine optimal prediction thresholds for all $\hat{o}_i$, aligning the prediction rates with the matching probabilities. This calibration is performed on the fly by tracking matching frequencies and computing the corresponding quantiles using a streaming algorithm. The proposed calibration algorithm is outlined in Algorithm 1.

---

**Algorithm 1** On-line calibration

---

**Input:** batches $b_n, n = \overline{1, N}$ with size $B$, momentum $m$.
**Output:** prediction thresholds $th_k, k = \overline{1, K}$.
  1: $r_k \leftarrow 1, k = \overline{1, K}$          ▷ *Initialize head matching rates*
  2: $th_k \leftarrow 0, k = \overline{1, K}$          ▷ *and thresholds*
  3: **for** $n = 1$ **to** $N$ :          ▷ *Process each batch*
  4:     $\text{matched}_k \leftarrow 0, k = \overline{1, K}$
  5:     $\text{scores}_k \leftarrow \emptyset, k = \overline{1, K}$
  6:     **for** $(y, \hat{y})$ **in** $b_n$ :          ▷ *Compute statistics for batch elements*
  7:        $\sigma \leftarrow$ optimal matching for $(y, \hat{y})$ given by Eq.4.
  8:        $\text{matched}_k \leftarrow \text{matched}_k + \text{I}\left[k \in \sigma\right], k = \overline{1, K}$    ▷ I *is the indicator function*
  9:        $\text{scores}_k \leftarrow \text{scores}_k \bigcup \{\hat{o}_k\}, k = \overline{1, K}$
 10:    $r_k \leftarrow (1 - m)r_k + m \, \text{matched}_k/B, k = \overline{1, K}$    ▷ *Update matching rates*
 11:    $th_k \leftarrow (1 - m)th_k + m \, \text{Q}(r_k, \text{scores}_k), k = \overline{1, K}$    ▷ *and thresholds.* Q *is the quantile function*

---

### 4.5 HYPERPARAMETER SELECTION

DeTPP relies on two key hyperparameters: the maximum number of predictions $K$ and the horizon $H$. We set the horizon $H$ to match that of the T-mAP metric, ensuring consistency in evaluation. The selected value is also sufficient to include the necessary number of events for calculating the OTD metric. The selection of $K$ typically requires tuning for each dataset. It controls the maximum number of predictions, and we recommend setting $K$ to approximately four times the average se-

quence length. In our experiments, $K$ values ranged between 32 and 64, depending on the dataset's characteristics. The value of $\lambda$ from Eq. 7 was set to 4 in all experiments.

Additionally, we found it beneficial to adjust the weight of each loss function during alignment to accommodate the number of model outputs, dataset classes, and the average time step. In practice, the optimal weight for $\mathcal{L}_{\mathrm{BCE}}$ is typically around eight times larger than the weights for the label and timestamp losses. For datasets with larger time steps, such as Retweet, the MAE loss weight should be reduced accordingly.

The exact hyperparameter values used in our experiments are provided in Appendix B.

## 5 EXPERIMENTS

We conducted a series of experiments using the HoTPP benchmark (Karpukhin et al., 2024) to assess the performance of DeTPP against several widely used MTPP models. Specifically, we compare DeTPP to Intensity-Free (Shchur et al., 2019), intensity-based RMTPP (Du et al., 2016) and NHP (Mei & Eisner, 2017) approaches, ODE-RNN (Rubanova et al., 2019), the transformer-based At-tNHP model Yang et al. (2022), and the long-horizon rescoring method HYPRO (Xue et al., 2022). We employ a GRU recurrent neural network Cho et al. (2014) as the backbone for the DeTPP model, similar to IFTPP and RMTPP.

The datasets employed in this study include Retweet (Zhao et al., 2015), Amazon (Jianmo, 2018), StackOverflow (Jure, 2014), MIMIC-IV (Johnson et al., 2023), and Transactions (AI-Academy for teens, 2021), which represent a diverse range of domains and scales, as summarized in Table 1.

Table 1: Datasets statistics and evaluation parameters

| Dataset | Domain | Sequences | Events | Classes | Time unit | OTD steps | Horizon / Mean length |
|---------|--------|-----------|--------|---------|-----------|-----------|-----------------------|
| StackOverflow | Social. net. | 2k | 138k | 22 | Minute | 10 | 10 / 12.0 |
| Amazon | Social. net. | 9k | 403k | 16 | N/A | 5 | 10 / 14.8 |
| Retweet | Social. net. | 23k | 1.3M | 3 | Second | 10 | 180 / 14.7 |
| MIMIC-IV | Medical | 120k | 2.4M | 34 | Day | 5 | 28 / 6.6 |
| Transactions | Financial | 50k | 43.7M | 203 | Day | 5 | 7 / 9.0 |

MTPP models are typically evaluated based on their accuracy in predicting the next event (Xue et al., 2023). Time and type predictions are often assessed separately, with type prediction accuracy measured by error rates and time prediction accuracy evaluated using metrics like Mean Absolute Error (MAE) or Root Mean Squared Error (RMSE). Recent advancements have introduced metrics such as Optimal Transport Distance (OTD) (Mei et al., 2019) and Temporal mAP (T-mAP) (Karpukhin et al., 2024), which are designed to evaluate long-horizon predictions by comparing predicted sequences to ground truth sequences within a specified horizon. In this work, we use all mentioned metrics to assess the performance of our proposed method. We also measure the diversity of predictions by using the predicted label distribution entropy. For further details on metric computation, please refer to Appendix E. Ablation studies are presented in Appendix F.

### 5.1 LONG-HORIZON EVENTS FORECASTING

We evaluate long-horizon prediction performance using Optimal Transport Distance (OTD) and Temporal mean Average Precision (T-mAP). As shown in Table 2, DeTPP significantly outperforms popular MTPP approaches, with the only exception being the OTD metric on the MIMIC-IV dataset, where IFTPP ranks first and DeTPP second. Overall, DeTPP achieves state-of-the-art performance in 9 out of 10 comparisons. The high T-mAP scores of DeTPP can be linked to its training objective, which utilizes matching, similar to T-mAP. However, DeTPP consistently improves the OTD metric, suggesting that its training process enhances overall model performance rather than merely optimizing for a single evaluation criterion.

Table 2: Evaluation results in the long-horizon prediction task. The best result is shown in bold. Mean and STD values of 5 runs with different random seeds are reported.

| Model | Metrics (OTD / T-mAP) | | | | |
|-------|------------------------|---|---|---|---|
| | StackOverflow | Amazon | Retweet | MIMIC-IV | Transactions |
| IFTPP | 13.64 / 8.31% ±0.05 / ±0.50% | 6.52 / 22.56% ±0.05 / ±0.52% | 172.7 / 31.75% ±4.4 / ±4.44% | **11.53** / 21.67% ±0.01 / ±0.21% | 6.90 / 5.88% ±0.01 / ±0.13% |
| RMTPP | 13.17 / 12.72% ±0.05 / ±0.16% | 6.57 / 20.06% ±0.03 / ±0.33% | 166.7 / 44.74% ±3.3 / ±0.94% | 13.71 / 21.08% ±0.03 / ±0.29% | 6.88 / 6.69% ±0.01 / ±0.12% |
| NHP | 13.24 / 11.96% ±0.02 / ±0.40% | 9.02 / 26.29% ±0.35 / ±0.55% | 165.8 / 45.07% ±1.6 / ±0.34% | 18.60 / 7.32% ±0.19 / ±1.33% | 6.98 / 5.61% ±0.01 / ±0.05% |
| AttNHP | 13.30 / 11.13% ±0.02 / ±0.32% | 7.30 / 14.62% ±0.06 / ±0.80% | 171.6 / 25.85% ±1.0 / ±1.08% | 14.68 / 22.46% ±0.08 / ±0.40% | 7.50 / 1.48% N/A / N/A |
| ODE | 13.27 / 10.52% ±0.03 / ±0.23% | 9.46 / 22.96% ±0.08 / ±0.61% | 165.3 / 44.81% ±0.5 / ±0.69% | 14.74 / 15.18% ±0.34 / ±0.15% | 6.97 / 5.52% ±0.01 / ±0.13% |
| HYPRO | 13.26 / 14.69% N/A / N/A | 6.61 / 20.53% N/A / N/A | 170.7 / 46.99% N/A / N/A | 14.87 / 16.77% N/A / N/A | 7.05 / 7.05% N/A / N/A |
| DeTPP | **12.14 / 22.72%** ±0.04 / ±0.32% | **5.98 / 37.20%** ±0.04 / ±0.06% | **132.9 / 57.93%** ±0.7 / ±0.33% | 12.95 / **30.35%** ±0.32 / ±0.25% | **6.70 / 9.26%** ±0.03 / ±0.09% |

The suboptimal performance of DeTPP on the MIMIC-IV dataset in terms of OTD warrants further discussion. As illustrated in Fig. 4a, MIMIC-IV includes numerous events with near-zero times-tamps. These events, however, follow a natural sequence (e.g., admission, laboratory tests, ICU stay start, treatment, ICU stay end). Consequently, for MIMIC-IV, precise ordering of events becomes more important than accurate time predictions.

To highlight the impact of ordering, consider a simple dataset consisting of a single sequence with 16 events, each having a timestamp of zero and labels forming an ordered sequence of integers from 0 to 15. In this dataset, timestamps provide no information about the event order. OTD, calculated for sequences of length 5, requires the model to correctly predict the first 5 events to achieve a low score. As shown in Fig. 4b, only IFTPP and RMTPP attain a low OTD score, while other methods, including those with high T-mAP scores, perform significantly worse. This outcome is expected for DeTPP, as its event ordering relies entirely on the predicted timestamps. Interestingly, NHP and ODE-based approaches exhibit similar behavior. This can be attributed to the NHP objective, which models the arrival times of each event class independently, leading to random ordering when timestamps are identical.

From this experiment, we conclude that most TPP methods struggle with ordering events that share equal timestamps. However, such problems fall beyond the scope of MTPP modeling, as outlined in Section 2.

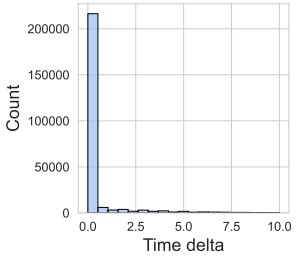
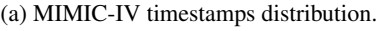

(a) MIMIC-IV timestamps distribution.

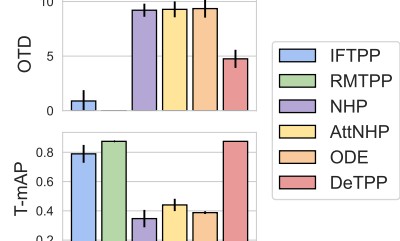

(b) Prediction quality on the Toy dataset with constant timestamps.

Figure 4: Additional experiments demonstrating specifics of the OTD metric on MIMIC-IV.

## 5.2 NEXT EVENT PREDICTION

Marked temporal point processes are usually evaluated based on the quality of next-event prediction. We measured the next-event type error rate and mean absolute time error across various datasets, with the results presented in Fig. 5. DeTPP achieves state-of-the art results in all comparisons and significantly reduces error on the Transactions dataset, the only dataset where the difference between top methods is significant. This improvement highlights DeTPP's potential to scale effectively for more complex problems involving a large number of classes and events. We conclude that DeTPP, although primarily designed for long-horizon prediction, also achieves high-quality performance in the next-event prediction task.

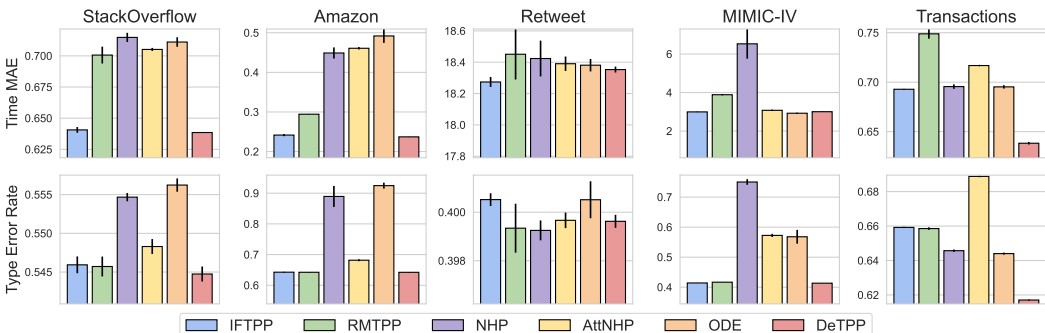

Figure 5: The next event prediction error.

## 5.3 PREDICTIONS DIVERSITY

As we qualitatively demonstrated in Section 3, autoregressive methods tend to produce repetitive outputs. In this section, we provide additional quantitative results that further highlight the low output diversity of traditional approaches.

A popular way to make diverse predictions is to adjust temperature during sampling. When the temperature approaches zero, the model predicts the label with the maximum output probability, as previous experiments did. When the temperature increases, the model samples from the uniform label distribution, leading to the highest possible entropy. We, therefore, need to study the impact of the temperature on both the predictions diversity and long-horizon quality. We do this by varying temperature from 0 to 10 and measuring the average entropy of predicted event types within the horizon. We also evaluate the long-horizon prediction quality regarding OTD, as T-mAP doesn't depend on temperature. The results from Fig. 6 show, that DeTPP achieves the optimal balance between diversity and prediction quality on 4 out of 5 datasets. Certain methods benefit from sampling-based approaches over maximum probability predictions; however, they fail to achieve the prediction quality of DeTPP. Further details on DeTPP's output diversity can be found in Appendix D.

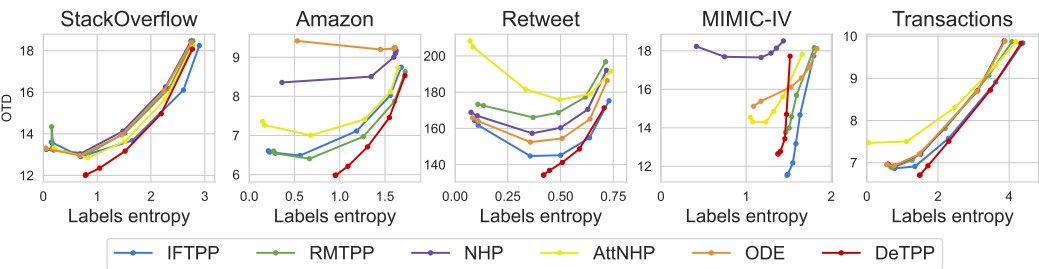

Figure 6: The relation between OTD and predictions diversity for varying sampling temperature values. The optimal quality corresponds to the bottom right corner, combining low error and high diversity.

## 5.4 TRAINING AND INFERENCE SPEED

A key practical aspect of the model is its computational efficiency. Fig. 7 presents a comparison of the training and inference times in terms of Requests Per Second (RPS), i.e., the number of processed batch elements[2]. The results show that DeTPP has a moderate training time while being the fastest method during inference across all datasets except Transactions, where it is slightly slower than IFTPP due to the computational cost associated with the prediction head. We therefore conclude that DeTPP is among the most computationally efficient methods in the field.

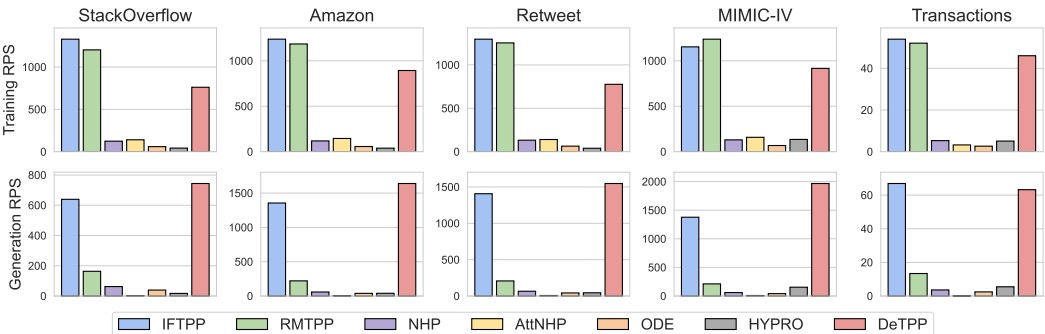

Figure 7: Computation speed measured as Requests Per Second (RPS) during training and sequence generation.

## 5.5 THE NUMBER OF HEADS

An important hyperparameter of DeTPP is the number $K$ of prediction heads. We analyzed the quality of the model depending on $K$ according to the T-mAP metric, as it, unlike OTD, considers the whole horizon. The results are presented in Fig. 8. It can be seen that the number of heads is an important hyperparameter that must be tuned individually for each dataset. In most cases, however, the optimal value lies between 32 and 64. It is also worth noting that the dependency is not monotonic, and the large $K$ might be suboptimal. On the MIMIC-IV and Transactions datasets, the value of $K$ has little impact on performance within a specified range, so we reduced the number of heads to speed up training.

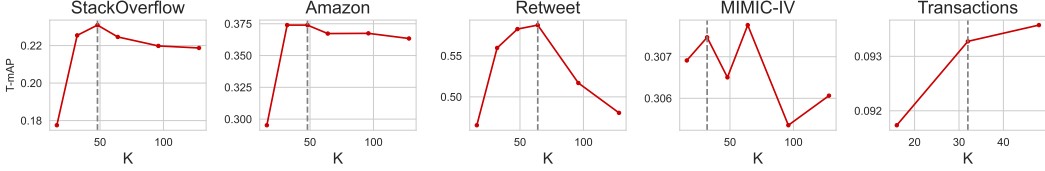

Figure 8: The dependency of T-mAP on the DeTPP hyperparameter $K$. The vertical line indicates the selected value.

## 5.6 HEADS SPECIALIZATION

To provide greater intuition on the functioning of DeTPP, we conducted an additional analysis of the outputs generated by the model's prediction heads. During training, each prediction head is matched to its closest corresponding ground truth event, leading to a gradual and persistent specialization. This specialization is illustrated in Fig. 9, where we present per-head output statistics. As shown, each prediction head becomes responsible for a specific time interval and a subset of event classes. Notably, the time intervals tend to be short in relation to the overall horizon length, while the number of assigned event classes can vary from 1 to 10. From this, we infer that the specialization of the prediction heads is driven more by temporal factors than by specific event types.

---

[2]The experiment was done with a single Nvidia RTX 4060 GPU.

In the Retweet dataset, approximately half of the heads focus on events occurring in the near future, leaving the later parts of the horizon largely uncovered. This suggests that the model naturally excludes rare and distant events, which are challenging to predict accurately. This behavior highlights DeTPP's inherent flexibility in adapting to the underlying data distribution without additional modifications.

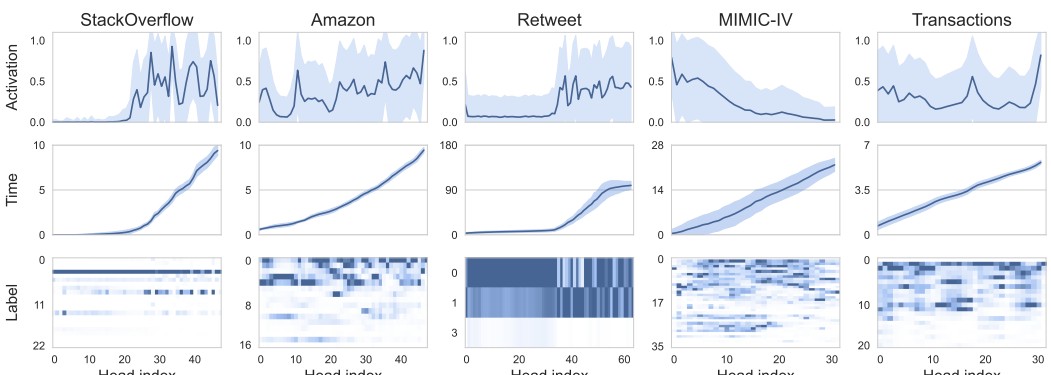

Figure 9: Output statistics across $K$ model heads. Head indices are sorted according to the mean predicted timestamp. The number of classes for Transactions is limited to 20 for clarity.

## 6 LIMITATIONS AND FUTURE WORK

While DeTPP effectively addresses both next-event and long-horizon prediction tasks, it has certain limitations. First, DeTPP relies on a fixed horizon size, $H$. In this study, we selected $H$ based on the hyperparameters of the OTD and T-mAP metrics. A change in the evaluation metric typically requires adjusting DeTPP's parameters. A potential direction for future research could explore combining horizon-based and autoregressive inference, enabling DeTPP to make predictions across arbitrary horizons without requiring modifications to the model.

Another limitation of DeTPP is its independent prediction of events within the horizon. Future research could investigate integrating rescoring techniques, such as those in Xue et al. (2022), or applying beam search to improve DeTPP's predictive performance. Additionally, better time modeling might be achieved by integrating intensity-based approaches, like NHP or RMTPP, with DeTPP, offering another promising direction for research.

Furthermore, some techniques from DeTPP could be adapted for object detection in computer vision Carion et al. (2020). DeTPP introduces a probabilistic framework that unifies different loss functions, using the same objective during both matching and backpropagation, which enhances optimization robustness. Notably, DeTPP employs a presence score during matching, which, as shown in our ablation studies, significantly improves the model performance on most datasets.

## 7 CONCLUSION

In this work, we introduced DeTPP, a novel event prediction method that adapts a set-based training objective from object detection to address the challenges of long-horizon forecasting. Our experiments demonstrate that DeTPP effectively overcomes the limitations of traditional autoregressive approaches. Notably, DeTPP generates more diverse predictions, significantly enhancing long-horizon prediction accuracy. Moreover, DeTPP is computationally more efficient than most MTPP methods, as it predicts multiple future events in parallel. This approach paves the way for advancing the modeling of Marked Temporal Point Processes, offering new opportunities across a wide range of practical applications.

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

## A  ON-LINE CALIBRATION ALGORITHM

The description of the algorithm moved to the main text. This section will be removed in the camera-ready version of the paper.

## B  HYPERPARAMETERS

Table 3 provides details of the backbone architecture and training parameters. Training on smaller datasets typically involves a greater number of epochs. For the Transactions dataset, the backbone size was increased to accommodate its higher number of events and event types compared to the other datasets.

The number of prediction heads $K$ and matching weights in DeTPP were chosen using a Bayesian optimizer, with the validation T-mAP score as the objective function. The selected horizon is equal to that used in the T-mAP metric. The $\lambda$ weight was set to 4 in all cases. The final values are

Table 3: Backbone and training hyperparameters.

| Dataset | Num epochs | Max Seq. Len. | Label Emb. Size | Hidden Size | Head layers dimensions |
|---|---|---|---|---|---|
| Transactions | 30 | 1200 | 256 | 512 | $512 \rightarrow 256$ |
| MIMIC-IV | 30 | 64 | 16 | 64 | 64 |
| Retweet | 30 | 264 | 16 | 64 | 64 |
| Amazon | 60 | 94 | 32 | 64 | 64 |
| StackOverflow | 60 | 101 | 32 | 64 | 64 |

Table 4: DeTPP hyperparameters.

| Dataset | K | Horizon | $\lambda$ | Matching weights | | |
|---|---|---|---|---|---|---|
| | | | | Presence | Timestamps | Labels |
| Transactions | 32 | 7 | 4 | 4 | 0.47 | 0.57 |
| MIMIC-IV | 32 | 28 | 4 | 4 | 0.4 | 0.8 |
| Retweet | 64 | 180 | 4 | 4 | 0.15 | 0.6 |
| Amazon | 48 | 10 | 4 | 4 | 0.5 | 0.8 |
| StackOverflow | 48 | 10 | 4 | 4 | 2 | 1 |

presented in Table 4. Note, that only relative values of matching weights are important. Thus, DeTPP has 3 hyperparameters that must be tuned.

We also conducted an additional analysis to examine the dependency of DeTPP on the selected loss weights. The results for the Amazon, Retweet, and StackOverflow datasets are shown in Figure 10 and Figure 11. Most parameters are observed to be near their optimal values, with a few suboptimal exceptions. We retained the original values to ensure a fair comparison with the baselines, where hyperparameters were also automatically tuned using Bayesian optimization.

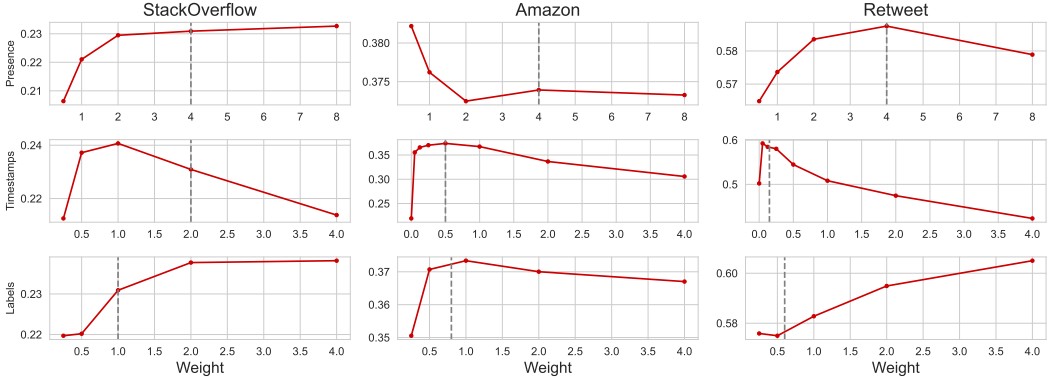

Figure 10: DeTPP T-mAP performance across varying loss weights. The vertical line indicates the value determined through Bayesian hyperparameter optimization.

## C  OTD ON MIMIC-IV

The section moved to the main text. Appendix will be removed in the camera-ready version of the paper.

## D  DeTPP PREDICTION DIVERSITY

In this section, we investigate the factors contributing to the improved output diversity achieved by DeTPP. Consider a toy dataset consisting of regular events with timestamps $0, 1, 2, 3, \ldots$ and random labels sampled from a categorical distribution with probabilities $(0.1, 0.2, 0.7)$, resulting in

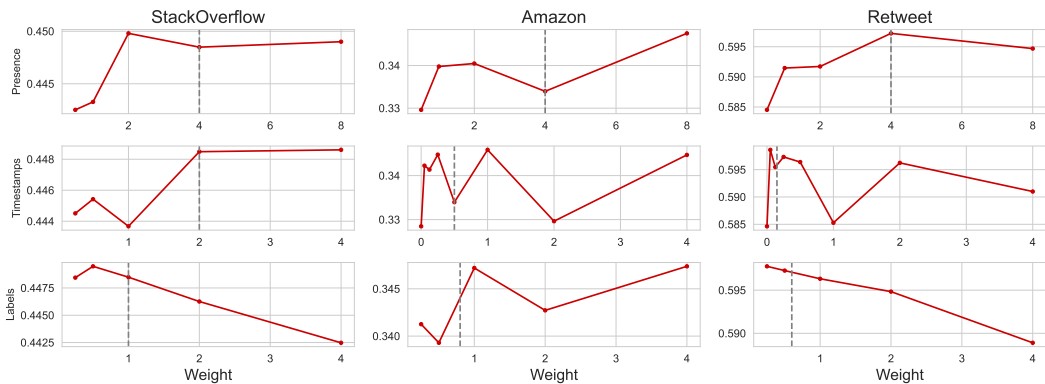

Figure 11: DeTPP accuracy across varying loss weights. The vertical line indicates the value determined through Bayesian hyperparameter optimization.

three possible labels. Suppose each label is sampled independently. In this scenario, historical data provides no information about future events, requiring the model to learn the prior distribution of events.

Since history offers no useful insights into the next event, autoregressive models tend to predict only the most probable event type (the third label), as illustrated in Figure 12. In contrast, DeTPP estimates the typical distribution of labels over the entire horizon. As a result, DeTPP's outputs include a significant proportion of events belonging to less frequent labels.

We conclude that DeTPP's improved prediction diversity stems from its ability to model the distribution of labels across the prediction horizon.

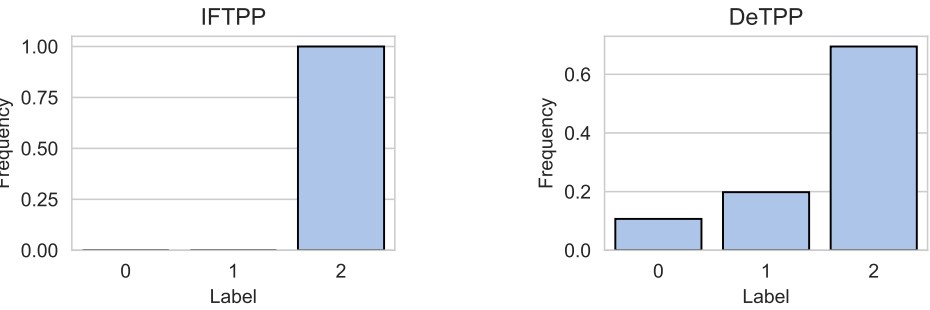

Figure 12: Predicted labels distribution on the toy dataset with independent labels.

## E  METRICS DESCRIPTION

In Section 5, we evaluate the models using both next-event prediction metrics and long-horizon prediction metrics. In this section, we provide detailed definitions of these metrics.

### E.1  NEXT-EVENT PREDICTION

Each event $y_i$ is defined by its timestamp $t_i$ and label $l_i$. The quality of the predicted timestamps is usually evaluated using the Mean Absolute Error (MAE) or Mean Squared Error (MSE). Since event sequences often contain many large time intervals, we choose MAE as the primary measure because it is more robust to outliers. The MAE for timestamps is defined as:

$$\text{MAE} = \frac{1}{N} \sum_{i=1}^{N} \left| t_i - \hat{t}_i \right|, \tag{8}$$

where $N$ is the total number of predictions, and $\hat{t}_i$ is the predicted timestamp for the $i$-th event. The error rate for labels is defined as the fraction of incorrectly predicted labels:

$$\text{ErrorRate} = \frac{1}{N}\sum_{i=1}^{N}\mathbb{I}(l_i \neq \hat{l}_i), \tag{9}$$

where $\mathbb{I}(\cdot)$ is the indicator function, equal to 1 if the argument is true and 0 otherwise, and $\hat{l}_i$ is the predicted label for the $i$-th event.

### E.2 LONG-HORIZON PREDICTION.

We evaluate long-horizon prediction using two metrics: Optimal Transport Distance (OTD) and Temporal mean Average Precision (T-mAP).

**Optimal Transport Distance (OTD).** OTD measures the discrepancy between predicted and ground truth sequences by computing an optimal matching between their events. For each pair of sequences, OTD extracts prefixes of a predefined length $K$, which is a hyperparameter typically set between 5 and 10 events. Given the prefixes $y = \{(t_i, l_i)\}_{i=1}^{K}$ from the ground truth and $\hat{y} = \{(\hat{t}_i, \hat{l}_i)\}_{i=1}^{K}$ from the predictions, OTD solves an optimal transport problem using a cost matrix $C^{\text{OTD}}$ of size $K \times K$, where the cost between the $i$-th predicted event and the $j$-th ground truth event is defined as:

$$C_{i,j}^{\text{OTD}} = \begin{cases} |\hat{t}_i - t_j|, & \text{if } \hat{l}_i = l_j, \\ 2\text{R}, & \text{if } \hat{l}_i \neq l_j, \end{cases} \tag{10}$$

where R represents the cost of deleting a ground truth event or inserting a predicted event, making the substitution cost 2R when labels do not match. The Optimal Transport Distance is then defined as the minimum total cost overall one-to-one assignments $\sigma$:

$$\text{OTD} = \min_{\sigma \in \mathcal{A}}\sum_{i=1}^{K} C_{i,\sigma(i)}^{\text{OTD}}, \tag{11}$$

where $\mathcal{A}$ represents all permutations of $\{1, 2, \ldots, K\}$.

**Temporal mean Average Precision (T-mAP).** T-mAP introduces two key parameters: the horizon length $H$ and the maximum allowed time error $\delta$. From each pair of predicted and ground truth sequences, T-mAP extracts subsequences that fall within the specified time horizon $H$. Unlike the Optimal Transport Distance (OTD), T-mAP compares sequences of variable lengths. Additionally, T-mAP evaluates the predicted probability distribution of labels $\hat{p}(l)$ rather than the hard labels, which typically correspond to the label with the highest probability. T-mAP computes the Average Precision (AP) for each class, which is equivalent to the area under the precision-recall curve. The AP values for all classes are then averaged with equal weights, a process commonly referred to as macro averaging.

To compute T-mAP for a specific class $l$, consider some decision threshold $\tau$. T-mAP selects a subset of predicted events with probability $\hat{p}(l)$ exceeding the threshold $\tau$. A prediction $\hat{y}_i = (\hat{t}_i, l)$ is considered a true positive (TP) if there exists a ground truth event $y_j = (t_j, l)$ such that $|\hat{t}_i - t_j| \leq \delta$. Each ground truth event can be matched to at most one prediction, so the number of true positives does not exceed the total number of targets. Predictions not matched to any ground truth event are classified as false positives (FP), while unmatched ground truth events are considered false negatives (FN). For a given threshold $\tau$, precision and recall can then be calculated as:

$$\text{Precision} = \frac{\text{TP}}{\text{TP} + \text{FP}},$$

Table 5: Ablation studies results. The best result is shown in bold. Mean and STD values of 5 runs with different random seeds are reported.

| Model | Metrics (OTD / T-mAP) | | | | |
|---|---|---|---|---|---|
| | StackOverflow | Amazon | Retweet | MIMIC-IV | Transactions |
| DeTPP | 12.14 / 22.72% | 5.98 / **37.20%** | 132.9 / **57.93%** | 12.95 / 30.35% | 6.70 / 9.26% |
| | ±0.04 / ±0.32% | ±0.04 / ±0.06% | ±0.7 / ±0.33% | ±0.32 / ±0.25% | ±0.03 / ±0.09% |
| Pairwise loss | 13.51 / 11.42% | 6.68 / 22.57% | 167.9 / 34.73% | 13.18 / 22.30% | 7.19 / 4.43% |
| | ±0.06 / ±0.78% | ±0.01 / ±0.07% | ±2.8 / ±5.11% | ±0.05 / ±0.03% | ±0.00 / ±0.16% |
| DeTR matching | 13.62 / 18.08% | 6.39 / 36.38% | 193.2 / 48.96% | **12.52** / **33.62%** | 6.88 / **9.32%** |
| | ±0.04 / ±0.27% | ±0.02 / ±0.25% | ±1.5 / ±0.28% | ±0.42 / ±0.10% | ±0.02 / ±0.07% |
| $\lambda = 0$ | **12.06** / **23.11%** | 5.98 / 37.18% | 134.4 / 57.37% | 12.85 / 30.63% | 6.66 / 9.17% |
| | ±0.04 / ±0.08% | ±0.01 / ±0.07% | ±0.8 / ±0.67% | ±0.26 / ±0.14% | ±0.03 / ±0.11% |
| Without cond. head | 12.27 / 19.87% | **5.97** / 37.09% | **131.2** / 55.37% | 13.32 / 31.13% | **6.64** / 9.03% |
| | ±0.02 / ±0.26% | ±0.06 / ±0.09% | ±0.4 / ±0.40% | ±0.11 / ±0.17% | ±0.03 / ±0.23% |

$$\text{Recall} = \frac{\text{TP}}{\text{TP} + \text{FN}}.$$

By varying the threshold $\tau$, a precision-recall curve is generated, and the Average Precision (AP) is computed as the area under this curve. The final T-mAP score is obtained by averaging the AP values across all classes:

$$\text{T-mAP} = \frac{1}{L} \sum_{l=1}^{L} \text{AP}(l),$$

where $L$ represents the total number of classes.

# F   ABLATION STUDIES

## F.1   MATCHING LOSS

DeTPP involves two modifications compared to previous methods: a parallel prediction of multiple future events and a novel loss function. To study the impact of these modifications, we evaluated DeTPP with a simple pairwise loss. Unlike matching, this loss is computed between events on corresponding positions in the prediction and ground truth. As shown in Table 5, pairwise loss demonstrates a low long-horizon prediction quality, even worse than a simple IFTPP. A possible reason for this failure can be incorrect alignment between predictions and ground truth, leading to noisy gradients and reduced prediction quality. We therefore conclude that our matching loss is an essential part of the training pipeline.

## F.2   THE EFFECT OF $\lambda$

In Equation 7, we introduced an extension of the loss function with the next-event loss, controlled by $\lambda$. We compare DeTPP with and without next-event loss by setting $\lambda$ to zero in the latter case. According to the results from Table 5, the difference between the long-horizon prediction quality of DeTPP and DeTPP with $\lambda = 0$ is not significant. We therefore conclude that the next-event loss doesn't contribute to long-horizon prediction improvements.

## F.3   CONDITIONAL HEAD

We introduced a conditional head architecture in Section 4.3. In Table 5, we present evaluation results for a model with a simple feed-forward head. The resulting approach without conditional head shows slightly better OTD scores and slightly worse T-mAP values. At the same time, the feed-forward model has more model parameters. For example, it contains 140082 trainable parameters for the Retweet dataset, while conditional head is roughly half of that size – 71543. We, therefore, preferred conditional heads due to their low memory usage and better-scaling properties with the growing number of prediction heads.

### F.4 PRESENCE MATCHING

One of the fundamental differences between DeTPP and the DeTR method from object detection Carion et al. (2020) is the usage of the presence score for matching, as stated in Equation 6 and Equation 4. The usage of the presence score allows the model to select a subset of its heads depending on historical data, better adapting to a particular sequence of events during training. We conducted an additional study and measured the model's performance with $\mathcal{L}_{\text{BCE}}$ excluded during matching. According to the results from Table 5, the model trained with DeTR matching is inferior to DeTPP with a large margin on small datasets while giving moderate or no improvement in other cases. We, therefore, conclude that matching with the presence score improves the stability of the model among different dataset sizes.

