# OpenReview forum: "Leveraging Object Detection for Diverse and Accurate Long-Horizon Events Forecasting"
_ICLR.cc/2025/Conference — Submitted to ICLR 2025_

### Official Review · Reviewer_TSUL · 2024-10-27

**Soundness:** 2
**Presentation:** 1
**Contribution:** 1
**Rating:** 3
**Confidence:** 3

**Summary:**

This paper highlights key limitations of autoregressive models in long-horizon prediction, including error accumulation over time, which results in repetitive or constant outputs, and limited inference parallelism due to dependency on previous predictions. To overcome these issues, the authors propose DeTPP, a novel model inspired by object detection techniques that can predict multiple future events in parallel. DeTPP introduces a matching loss function that bypasses some events and focuses instead on accurately predicting more reliable ones. This approach achieves state-of-the-art performance in long-horizon forecasting, outperforming both autoregressive and next-K models. Additionally, an extension that integrates elements of traditional methods with DeTPP enhances next-event prediction quality, especially on large datasets like Transactions

**Strengths:**

The paper archives competitive results on multiple datasets for event forecasting. Additionally, it shows improved inference speed over most of the evaluated datasets. The proposed method exhibits greater diversity in its predictions.

**Weaknesses:**

The paper has several significant issues regarding method novelty, presentation, and experimental design.

**1. Methodology**
- **Lack of Novelty**: The paper’s claimed contributions to method development are unclear, as it appears to simply modify next-K models (Karpukhin et al., 2024). Thus, the claim of addressing autoregressive limitations in long-horizon prediction may not be entirely valid. Limited inference parallelism is inherent to autoregressive models and doesn’t represent a distinct limitation of previous methods; replacing the autoregressive model naturally addresses this issue to some degree. I suggest the authors explain how their approach differs from or improves upon next-K models in addressing autoregressive limitations.
- **Object Detection Inspiration**: Although the authors claim that DeTPP (Detection-based Temporal Point Processes) is inspired by object detection techniques in computer vision, they don’t provide a clear rationale for why these techniques are beneficial in this context.
- **Method Description**: The method section is incomplete and lacks structure. It begins with “4.1 Probabilistic Event Model,” describing loss functions without first introducing input and output notations. Key details, like the neural network used, are missing. If the model only includes what is described in Section 4.3, what does “Backbone” represent in Figure 2? Furthermore, there’s no explicit mention of object detection inspiration within the methodology.

**2. Experimental Design**
- **Missing Ablations**: Several important ablation studies are missing, including:
  - The impact of the losses outlined in Sections 4.1 and 4.2.
  - The effect of adjusting the loss weights.
  - The influence of model architecture choices, such as alternative methods for combining Queries and Embeddings (e.g., cross-attention) as seen in Figure 3.
  - An ablation study on the number of queries.
- **Performance on Next-Event Prediction**: Section 5.2 points out that the model struggles with next-event prediction. Even with the addition of the IFTPP loss function, results (Figure 4) on datasets like StackOverflow, Retweet, and MIMIC-IV show little improvement over the IFTPP method, suggesting limited effectiveness in this regard.

**3. Inference Speed**
- **Variation in Requests Per Second (RPS)**: The Requests Per Second (RPS) for sequence generation varies across datasets, as shown in Figure 6. The proposed method is the fastest on all datasets except Transactions, which is slower than IFTPP. The authors attribute this to computational overhead related to the prediction head but don’t explain why this issue affects only the Transactions dataset. The authors could offer a more detailed explanation of why the computational overhead of the prediction head impacts the Transactions dataset differently than the other datasets.

**Questions:**

See weakness.

---

> ### Author Response · Authors · 2024-11-21
>
> We deeply appreciate your detailed review and the time you have dedicated to evaluating our work. In the responses below, we address your comments point by point, highlighting the changes made in the updated version of the manuscript and providing additional context where needed.
>
> **W 1.a. Lack of novelty.** Please refer to our general response. While inference speed and parallelism in Next-K approaches have been explored in previous works, our primary motivation, as outlined in Section 3, is to address how predictions are matched with ground truth within the loss function. To the best of our knowledge, this aspect is novel in the context of event sequences. Our contributions lead to the development of a new loss function specifically designed for training event sequence models, resulting in superior prediction quality. Overall, DeTPP represents a significant advancement and refinement of the Next-K approach.
>
> **W 1.b. Relation to object detection.** The objective of our work is not simply to apply object detection techniques to event sequences but to address specific challenges inherent to this domain. In Section 3, we highlight the limitations of autoregressive approaches and demonstrate the need for an improved loss function capable of correctly aligning predictions with ground truth events. We draw inspiration from the DeTR loss function [1], which addresses the "set prediction problem", while the backbone architecture itself is not the central focus of our study.
>
> Other widely used object detection methods, such as anchor-based approaches [2], offer an alternative perspective. Applying anchors to DeTPP would result in a model variant where each head predicts events for a predefined time interval in the future. However, our implementation provides greater flexibility. As demonstrated in Section 5.5, DeTPP automatically determines the target time intervals for each prediction head based on the data, eliminating the need for predefined anchors and enhancing adaptability to diverse datasets.
>
> **W 1.c Method description.** In the updated version of the paper, we have improved the introduction in Section 4 and revised the subsection titles to better align with Figure 2. The domain of event sequences is inherently complex, combining irregular time series with tabular (structured) data. To address this, we introduce DeTPP in a step-by-step manner, progressing from simple concepts to more complex ones.
>
> In Section 4.1, we begin by modeling individual events, establishing a foundation for understanding the core principles of our approach. We then extend this framework to sequences of events, focusing on the long-horizon prediction task. Finally, we describe additional enhancements and refinements to the method. DeTPP introduces novelty at all three levels, and we believe that this incremental approach enhances clarity and comprehension, as opposed to starting with a large and complex task from the outset.
>
> **W 1.c Backbone.** Thank you for pointing out the need for a clearer description of the backbone. DeTPP employs a single-layer GRU network, similar to IFTPP and RMTPP. We have included a backbone description in Section 5 of the updated version of the paper.
>
> [1] Carion N. et al. “End-to-end object detection with transformers”, ECCV, 2020
>
> [2] Ren S. et al. “Faster R-CNN: Towards real-time object detection with region proposal networks”, IEEE transactions on pattern analysis and machine intelligence, 2016

---

> > ### Author Response · Authors · 2024-11-21
> > **(continue)**
> >
> > **W 2.a.1 Loss ablations.** We have conducted ablation studies to examine the components of the loss function that significantly differ from DeTR. These studies, detailed in Section 5.6, include:
> >
> > (a) a comparison between the simple Next-K loss and our matching-based loss,
> >
> > (b) an evaluation of matching with and without the inclusion of the "no-object" class, and
> >
> > (c) the impact of adding IFTPP loss during training.
> >
> > Additionally, as part of the hyperparameter search, we experimented with using MSE for time loss instead of MAE and Focal loss in place of simple BCE. Our findings indicate that MAE loss and simple BCE consistently yield superior performance.
> >
> > **W 2.a.2 The effect of adjusting the loss weights.** Thank you for raising this question. We conducted experiments to analyze the impact of varying loss weights, with the results presented in Appendix B of the updated paper. The analysis shows that most parameters are close to their optimal values, with a few exceptions. To ensure a fair comparison with the baselines, we retained the current parameter values, which were determined using Bayesian optimization, consistent with the tuning approach for the baselines.
> >
> > **W 2.a.3 Alternative architectures.** The primary focus of our work is on developing a novel loss function, as discussed in Section 3, where we address the limitations of autoregressive losses and propose a solution to overcome them. Our architecture employs a GRU backbone, inherited from methods like IFTPP and RMTPP, to ensure compatibility with existing frameworks. The resulting method delivers impressive performance. As demonstrated in Section 5.4, Transformer architectures exhibit lower training and inference speed compared to RNNs, particularly during autoregressive inference, which involves multiple sequential steps. Given these considerations, we opted for RNNs due to their ability to achieve high prediction quality while maintaining faster computation speed. While further exploration of alternative architectures is an interesting direction, it lies beyond the scope of the research questions addressed in this study.
> >
> > **W 2.a.4 Number of queries.** The number of queries in our model corresponds to the number of prediction heads. We investigate the impact of this design choice in Section 5.6.5, where an ablation study evaluates the effect of varying the number of prediction heads.
> >
> > **W 2.2 Performance on Next-Event Prediction.** As indicated by the title, the primary focus of our work is long-horizon prediction. However, we also address the question of developing a universal model capable of handling both next-event and long-horizon prediction tasks. To this end, we propose DeTPP+, which demonstrates its ability to excel in both domains. DeTPP+ achieves significant improvements in long-horizon prediction while matching or surpassing state-of-the-art (SOTA) methods. Notably, it achieves a significant improvement on the largest Transactions dataset. Based on these results, we conclude that DeTPP+ is competitive with SOTA methods and, in some cases, significantly outperforms them in next-event prediction tasks.
> >
> > **W 3. Inference speed.** Details regarding the architecture are included in Appendix B of the updated version of the paper. These details reveal that the transaction models incorporate a greater number of parameters in the fully connected output layer. In DeTPP, the prediction head is applied to each query, making the size of the prediction head a more significant factor affecting inference speed than the backbone complexity. Conversely, DeTPP is less impacted by the complexity of the backbone compared to autoregressive approaches. These considerations explain why IFTPP demonstrates faster inference in the specific experiment, despite DeTPP’s overall efficiency.
> >
> > We sincerely hope that our responses and the revisions made to the manuscript address your concerns and clarify the contributions of our work. We kindly invite you to share your thoughts on our considerations and updates.

---

### Official Review · Reviewer_jgTG · 2024-10-30

**Soundness:** 3
**Presentation:** 2
**Contribution:** 2
**Rating:** 5
**Confidence:** 2

**Summary:**

Paper proposes a novel event forecasting method which predicts multiple events in parallel. Most tradtional methods predict events in sequences. This can lead to error propogration and other issues, like over-uniformity. The method claims to be inspired by transformer-based object detection methods. There are several enhancements/variants of the approach to address different limitations of the base model. The experimental results compare favourably with many SOTA methods.

**Strengths:**

1. Paper proposes a novel method to predict long horizon events in parallel and obtain good empirical results.
2. The loss function design is sound and logical.
3. From the experimental results, the proposed base method appears to have overcome some limitations of existing SOTA. In Table 2, the method outperforms 6 other methods 9 out of 10 comparisons (5 datasets with 2 metrics: OTD/T-mAP)
4. There are several enhancements/variants of the base method to address the limitations of the original model.

**Weaknesses:**

1. Paper is difficult to read for readers without much background in this problem. For example, Line 258-259, "We set the horizon H to align with that of the T-mAP metric, ensuring consistency in evaluation." It's not clear how a hyperparameter can be "aligned" with the metric. Does this mean that different H value were experimented on, and the H value is set based on the best T-mAP?

2. The enhancements/variants appears to be ad-hoc. This can be seen in Table 3. The empirical results are mixed for the different datasets. There is no clear advantage of one variant over the rest. Each variant appear to be empirical tweaks and not designed based on sound theorical principles.

3. (minor) The paper's main claim that the method is inspired by object detection method. But there is no single object detection approach. Object detection is still an open research problem and there are other competing methods, besides transformer-based approach. In fact, the paper only cited one reference (Carion et al, 2020). Further, the paper does not mention exactly which part of their proposed method is directly inspired by Carion et al. The reader has to read between the lines and be quite familiar with Carion et al paper to draw their inference of the inspiration.

**Questions:**

1. Please consider if all variants are necessary to demonstrate the strengths of the proposed approach. As of the current state of the paper, the variants are actually diluting the core contribution by introducing unnecessary tweaks and make the paper difficult to read and appreciate. I suggest the paper to introduce DeTPP+ as the base model and performs ablation study with ablated model like DeTPP.

2. The exclusion of the other variants can free up space to elaborate on the experimental setup design. In the current state of the paper, the Section on the various parameters values is too brief. The Calibration process also appears to be a key component of the proposal and should be left in the main paper, rather than in the Appendix.

**Details Of Ethics Concerns:**

No concern.

---

> ### Author Response · Authors · 2024-11-21
>
> We thank you for your thoughtful and comprehensive feedback. Your insights and questions have provided us with an opportunity to clarify and further strengthen our work. Below, we address each of your points in detail, highlighting the revisions and improvements made in response to your suggestions.
>
> **W1. The meaning of “aligned”.** In the updated version of the paper, we have clarified the meaning of “aligned” and provided a brief explanation of the evaluation metrics in Appendix E. Specifically, the T-mAP metric employs the horizon length H as a hyperparameter. To ensure consistency, we set the prediction horizon of DeTPP to match the horizon of T-mAP, as extending the horizon would not impact the evaluation results.
>
> **W2. The enhancements/variants appear to be ad-hoc.** DeTPP introduces several improvements and updates over DeTR [1] and other methods, each of which is thoroughly evaluated through ablation experiments. These experiments analyze the contribution of each component independently, ensuring a systematic approach. We also provide clear motivation for each enhancement. For instance, the motivation behind DeTPP+ is discussed in Section 5.2, where it is presented as a universal model capable of addressing both long-horizon and next-event prediction tasks. Similarly, the rationale for employing a conditional head (as compared to DeTPP-FF) is explained in Section 4.4, highlighting its role in significantly reducing the model's parameter count while maintaining performance. The inclusion of the “no-object” probability during matching is justified by its ability to unify matching and training objectives (Section 4.2), resulting in a consistent maximum likelihood formulation—an aspect absent in DeTR. While certain DeTPP variants show improved performance in specific scenarios, we have selected the variant that delivers the best overall quality.
>
> **W3. Relation to the general object detection field.** The objective of our work is not simply to apply object detection techniques to event sequences but to address specific challenges inherent to this domain. In Section 3, we highlight the limitations of autoregressive approaches and demonstrate the need for an improved loss function capable of correctly aligning predictions with ground truth events. We draw inspiration from the DeTR loss function, which addresses the "set prediction problem", while the backbone architecture itself is not the central focus of our study.
>
> Other widely used object detection methods, such as anchor-based approaches [2], offer an alternative perspective. Applying anchors to DeTPP would result in a model variant where each head predicts events for a predefined time interval in the future. However, our implementation provides greater flexibility. As demonstrated in Section 5.5, DeTPP automatically determines the target time intervals for each prediction head based on the data, eliminating the need for predefined anchors and enhancing adaptability to diverse datasets.
>
> [1] Carion N. et al. “End-to-end object detection with transformers”, ECCV, 2020
>
> [2] Ren S. et al. “Faster R-CNN: Towards real-time object detection with region proposal networks”, IEEE transactions on pattern analysis and machine intelligence, 2016

---

> > ### Author Response · Authors · 2024-11-21
> > **(continue)**
> >
> > **Q1. Remove/move ablations.** Ablation studies are a critical component of our experiments, as they evaluate and justify the contribution of each element independently. We believe retaining these studies in the main body of the paper is essential for a clear understanding of the key decisions that constitute DeTPP.
> >
> > DeTPP+ is a straightforward extension of DeTPP, designed to address a specific task: next-event prediction. Accordingly, we position DeTPP as the primary method, as it serves as the foundational approach for solving long-horizon prediction tasks. DeTPP+ is presented as a potential extension to handle next-event prediction scenarios.
> >
> > While other tasks, such as estimating the distributions of future events, could also be explored by adding additional classification heads to either DeTPP or DeTPP+, these directions extend beyond the current scope of our work and are therefore not included in this study.
> >
> > **Q2. Move calibration to the main text of the paper from appendix.** Technical details can be challenging to fully comprehend at first glance. Even pseudocode often provides only a high-level overview of the process, while the precise implementation may require dozens of lines of code and is best understood by reviewing the actual source (e.g., losses/detection.py:DetectionLoss.update_calibration_statistics).
> >
> > To balance clarity and accessibility, we have included a high-level explanation of the calibration process in the main text of the paper, offering readers an intuitive understanding of the method. Detailed technical specifics are retained in the appendix for those who may need to implement or modify the calibration algorithm themselves. This approach ensures that the main text remains accessible while providing all necessary information for more advanced exploration.
> >
> > We sincerely thank you once again for your thoughtful feedback and the opportunity to address your comments. We hope our responses have clarified the points you raised and demonstrated contributions of our work. We kindly invite you to share your thoughts on our considerations and, if appropriate, reflect these in your final evaluation.

---

> > ### Comment · Reviewer_jgTG · 2024-11-26
> >
> > W1. Thanks for your clarification.
> >
> > W2. While the motivations are mentioned in the paper, these appears to be posthoc efforts. Importantly, the variants are specific to certain scenarios as shown in the experimental results. In my view, it is better for these works to be supplementary and focus on the main model (DeTPP+) in the main text.
> >
> > W3. Thank you for highlighting that the main inspiration is drawn from the DeTR loss function. This was not clear in my reading. Please consider changing your paper's title and positioning from the "general object detection field inspiration" to the "DeTR loss function inspired".

---

> > > ### Author Response · Authors · 2024-11-27
> > >
> > > As per your suggestion, we have included only the extended DeTPP method in the main part of the paper, while moving the ablation studies to the Appendix. Additionally, the discussion on MIMIC-IV performance and the detailed calibration algorithm have been incorporated into the main text.
> > >
> > > We have also revised references to object detection throughout the paper, focusing primarily on the loss function. However, the analogy between object detection and MTPP remains valid, as we consider time instead of spatial dimensions. For this reason, we have decided to retain the current title.
> > >
> > > Thank you for your valuable feedback and thoughtful suggestions. We hope we have addressed all your concerns satisfactorily. If so, we kindly request you to update the final score.

---

### Official Review · Reviewer_E8N9 · 2024-11-03

**Soundness:** 2
**Presentation:** 2
**Contribution:** 2
**Rating:** 5
**Confidence:** 3

**Summary:**

Long-horizon event forecasting is critical in various domains such as retail, finance, healthcare, and social networks. Traditional methods like Marked Temporal Point Processes (MTPP) often rely on autoregressive models, which can converge to constant or repetitive outputs, limiting their effectiveness. To address these issues, we introduce DeTPP (Detection-based Temporal Point Processes), a novel approach inspired by object detection techniques from computer vision. DeTPP uses a unique matching-based loss function that prioritizes reliably predictable events, improving prediction accuracy and diversity. This method achieves up to a 77% relative improvement over existing MTPP and next-K methods and enhances next event prediction accuracy by up to 2.7% on a large transactional dataset. Notably, DeTPP is also among the fastest methods for inference.

**Strengths:**

The paper's innovative approach, DeTPP, addresses the unique challenges of long-horizon event forecasting by leveraging a novel matching-based loss function and parallel prediction. This results in significant improvements in prediction accuracy, diversity, and efficiency, making it a valuable tool for various real-world applications.

**Weaknesses:**

- DeTPP relies on a fixed horizon size, which is selected based on the hyperparameters of the OTD (Optimal Transport Distance) and T-mAP (Temporal Mean Average Precision) metrics. This fixed horizon size can be a limitation because changes in the evaluation metric typically require adjusting DeTPP’s parameters.

- The paper does not provide detailed explanations of the evaluation metrics used, such as T-mAP (Temporal Mean Average Precision). This lack of detail can make the paper feel disorganized and less accessible to readers who are not familiar with these metrics. Including clear definitions and explanations of the metrics would enhance the clarity and readability of the paper. For example, you could add a separate paragraph in Section 5 to introduce the metrics you used and the meanings of the abbreviations in your paper. You can consider using a comparison table for clarity.

- The writing style of the paper is somewhat informal, which can detract from its academic rigor and professionalism. The main text of the entire article does not reach 10 pages. The introduction in Section 2 (Related Work) lacks a coherent narrative. The evaluation part does not need to be a separate subsection; it can be integrated into the discussion of different models. In addition, the subheadings in Section 2 are not parallel in nature.

**Questions:**

See weakness.

---

> ### Author Response · Authors · 2024-11-21
>
> We sincerely thank you for your detailed and constructive feedback. Your comments and suggestions have provided valuable guidance for refining and improving our work. Below, we address each of your points thoroughly, referencing the corresponding updates made in the revised text.
>
> **W1. Fixed horizon size.** Indeed, the horizon length is a hyperparameter of the method, typically determined by specific business requirements in real-world applications. For instance, a bank may choose a horizon corresponding to a day, week, or month for modeling client behavior. In our experiments, we partially address the robustness of DeTPP to the selected horizon length. Specifically, OTD is computed based on the first 5 or 10 predicted events, depending on the dataset, which is generally 2 to 3 times shorter than the typical number of events within the full horizon. This approach ensures that OTD evaluates performance under a horizon different from T-mAP. Despite this, DeTPP consistently improves both metrics (with the exception of OTD on MIMIC-IV). Furthermore, DeTPP+ delivers high-quality results in next-event prediction tasks, achieving performance close to state-of-the-art (SOTA) methods on 4 out of 5 datasets and significantly improving prediction accuracy on the Transactions dataset. Thus, we conclude that DeTPP effectively optimizes T-mAP while also generalizing its quality improvements across other evaluation metrics.
>
> **W2. Metrics description.** We have included detailed metric descriptions in the Appendix E of the updated version of the paper.
>
> **W3.a. Informal language.** We have addressed the issue of informal language in the revised manuscript, ensuring a professional and scientific tone throughout (about 6 to 8 sentences).
>
> **W3.b. The main text of the entire article does not reach 10 pages.** The main text currently does not exceed 10 pages, leaving room for potential updates during the rebuttal phase. However, this approach aligns with the recommendations in the ICLR 2025 CFP: “We encourage authors to be crisp in their writing by submitting papers with 9 pages of main text. We recommend that authors only use the longer page limit in order to include larger and more detailed figures.”
>
> **W3.c. The introduction in Section 2 (Related Work) lacks a coherent narrative.** The introduction in Section 2 (Related Work) was intentionally designed as a topic-based reference to facilitate easy access to relevant citations when needed. In response to your feedback, we have simplified this section in the updated version of the paper by reducing the nesting level, improving its readability and coherence.
>
> **W3.d. The evaluation part does not need to be a separate subsection; it can be integrated into the discussion of different models.** Thank you for this valuable suggestion. In response, we have integrated the list of considered metrics into the experiments section 5, ensuring a more seamless flow of information. Please refer to the updated version of the paper.
>
> We hope that our responses and the updates to the manuscript address your concerns and provide clarity on the key aspects of our work. We greatly value your feedback and would appreciate it if you could share your thoughts on our considerations.

---

> ### Author Response · Authors · 2024-11-29
> **Discussion**
>
> As the discussion period is coming to a close, we kindly ask if we have adequately addressed all your questions and concerns. If our responses are satisfactory, we would greatly appreciate it if you could consider adjusting the final score accordingly. Thank you for your time and thoughtful feedback!

---

### Official Review · Reviewer_ffVJ · 2024-11-05

**Soundness:** 3
**Presentation:** 2
**Contribution:** 1
**Rating:** 3
**Confidence:** 4

**Summary:**

The authors aim to address the issues of wrong matching and repetitive outputs in long-horizon events forecasting. This work transfers some ideas and architectures from the field of object detection to the events forecasting, achieving superior results. The authors have open-sourced their code, which enhances the reproducibility and data reliability of this work.

**Strengths:**

1. The authors have leveraged some advantages of DETR and adapted it to the events forecasting, addressing certain shortcomings of autoregressive methods and Next-K approaches.
2. The authors have open-sourced their code, which enhances the reproducibility of the paper.

**Weaknesses:**

1. Lack of innovation. The authors have adopted the Hungarian matching from DETR with minimal modifications and made few targeted improvements (see Q.1, Q.2).
2. There is a lack of comparison with some advanced methods. Certain recent works, such as ContiFormer$^{[1]}$, have not been mentioned or compared.
<!-- 3. Insufficient evaluation. Although the authors state in Section 2.3 (Line 125-126), "In this work, we use all mentioned metrics to assess the performance of our proposed method," they only utilize OTD and T-mAP. They do not employ MAE and MSE, both of which are also widely used metrics. -->

[1] Chen, Yuqi, et al. "Contiformer: Continuous-time transformer for irregular time series modeling."

**Questions:**

1. The authors emphasize that an important difference between their method and DETR$^{[2]}$ is the introduction of alignment loss ($L_{BSE}$, Eq.(4)); however, this alignment loss does not seem significantly different from the first term $-log(\hat{p}_{\hat{\sigma}(i)}(c_i))$ in Eq.(2) of DETR. The main distinction is that DETR and its variants$^{[3,4]}$ treat "no object" as a special class, handling it equivalently to the trivial class when predicting logits. In contrast, this work separates "no event" from trivial events. If the authors consider this operation to be a key improvement, they should provide justification and corresponding experiments.

2. The authors use binary cross-entropy loss as the alignment loss, but many DETR-like methods$^{[3,4,5,6]}$ consider Focal Loss$^{[7]}$ to be more suitable because, during the decoding process, the number of positive samples is typically much smaller than that of negative samples. Using BCE loss may lead the model to be more inclined to classify samples as negative. Why was Focal Loss not used in this work? Generally, what is the ratio of positive to negative samples among the K predictions of this method?

3. What does the "conservative probability estimation" in Section 4.4 refer to? Does it mean that the probability of classifying samples as negative is higher?

4. The most recent method compared to this work is from 2022. Why is there no comparison with the latest methods, such as ContiFormer$^{[1]}$?

5. Figure 1(c) requires more clarification. What do the different shapes and colors represent? In the three rows of legends for each method, what does each row represent—ground truth, predictions, or do all three rows together form a single output?

6. The authors highlight the generation of more diverse outputs as an advantage and provide some qualitative analysis. However, a more in-depth theoretical explanation of why the DeTPP loss enables the generation of diverse outputs is needed.

[2] Carion, Nicolas, et al. "End-to-end object detection with transformers." European conference on computer vision. Cham: Springer International Publishing, 2020.
[3] Zhang, Hao, et al. "Dino: Detr with improved denoising anchor boxes for end-to-end object detection.".
[4] Zhu, Xizhou, et al. "Deformable detr: Deformable transformers for end-to-end object detection."
[5] Shi, Dahu, et al. "End-to-end multi-person pose estimation with transformers." Proceedings of the IEEE/CVF Conference on Computer Vision and Pattern Recognition. 2022.
[6] Yang, Jie, et al. "Explicit box detection unifies end-to-end multi-person pose estimation."
[7] Ross, T-YLPG, and G. K. H. P. Dollár. "Focal loss for dense object detection." proceedings of the IEEE conference on computer vision and pattern recognition. 2017.

---

> ### Author Response · Authors · 2024-11-21
>
> We thank you for your thoughtful and detailed feedback. Your comments have provided valuable insights and have given us the opportunity to clarify and strengthen our work. Below, we address each of your points thoroughly, referencing updates made in the revised paper and providing additional explanations where necessary.
>
> **W1: Lack of innovation.** We have addressed the novelty of our work comprehensively in the general response above (Novelty).
>
> **W2.a: Recent works such as ContiFormer.**
>
> Please refer to the general answer (Experiments and Baselines).
>
> **W2.b: Widely used metrics.**
>
> In Section 5.2, we report MAE for timestamps and error rates for labels. DeTPP+ achieves state-of-the-art results in these metrics across most datasets, with a significant advantage on the Transactions dataset.
>
> **Q1: Matching difference from DeTR.** In DeTR, the matching objective (Eq. 1) and the training objective (Eq. 2) are distinct. Specifically, while the "no-object" class contributes to the training objective in Eq. 2, it does not influence the matching process in Eq. 1. In contrast, our approach unifies these objectives by incorporating the probability of the "no-object" class directly into the matching process. This distinction sets our method apart, enabling a more coherent optimization framework. As demonstrated in our ablation studies (Section 5.6.4), this unified matching algorithm yields significant improvements on the Amazon, Retweet, and StackOverflow datasets, highlighting its effectiveness.
>
> **Q2: Focal loss.** We conducted experiments with Focal Loss (the code is available at losses/common.py:BinaryCrossEntropyLoss), using an automated hyperparameter search. However, the results of this search led to the removal of focal loss from the model, as it did not provide significant benefits. Unlike typical object detection tasks, our problem does not exhibit extreme class imbalance. The head matching rates for our datasets further support this observation: Amazon (38%), Retweet (22%), StackOverflow (25%), MIMIC (25%), and Transactions (28%).
>
> **Q3: “conservative probability estimates”.** We have clarified this point in the updated text (Section 4.3). The issue arises because probabilities trained with cross-entropy loss often converge close to prior probabilities when the fraction of classification errors is high. While the predicted probabilities do reflect model confidence—being higher for expected events and lower otherwise—they are typically below 50%. Using a fixed threshold of 0.5 in such cases would result in a low number of predictions and excessively short sequences. To address this, we calibrate the predicted probabilities to align with the actual frequency of events, ensuring more accurate and meaningful predictions.
>
> **Q4: Recent works, such as ContiFormer.**
>
> Please refer to the general answer (Experiments and Baselines).
>
> **Q5: Figure 1(c).** We have clarified the caption in the updated version of the paper. The figure depicts three predicted sequences from the Amazon dataset for three methods: IFTPP, IFTPP-K, and DeTPP. To simplify the visualization, we omitted detailed timestamps and focused solely on predicted labels. These sequences were directly extracted from model predictions during evaluation. Each predicted event type is represented by a unique combination of color and shape, corresponding to one of the 16 classes in the Amazon dataset. The qualitative results clearly illustrate that DeTPP predicts six distinct event types, while the other methods predict only two, often resulting in constant or repetitive patterns. A comprehensive quantitative analysis of prediction diversity is provided in Section 5.3.
>
> **Q6: Why DeTPP provide more diverse outputs?** Thank you for raising this question. To address it, we conducted an additional experiment using a toy dataset, with results presented in Appendix D of the updated paper. The toy dataset consists of random and independent events, where labels follow a categorical distribution with probabilities (0.1, 0.2, 0.7). Since historical observations contain no useful information, autoregressive approaches consistently predict the third event, corresponding to the maximum prior probability. In contrast, DeTPP models the full distribution of events over the prediction horizon. This enables it to predict a significant proportion of events of all types. This toy example highlights DeTPP’s ability to provide more diverse predictions by effectively modeling the entire event distribution over the horizon.
>
> We sincerely hope that our responses and the updates to the manuscript address your concerns and provide clarity regarding the contributions of our work. We kindly invite you to share your thoughts on our considerations and welcome any further feedback.

---

> ### Author Response · Authors · 2024-11-29
> **Discussion**
>
> As the discussion period is coming to a close, we kindly ask if we have adequately addressed all your questions and concerns. If our responses are satisfactory, we would greatly appreciate it if you could consider adjusting the final score accordingly. Thank you for your time and thoughtful feedback!

---

### Author Response · Authors · 2024-11-21
**General response**

We sincerely thank the reviewers for their thoughtful feedback and valuable suggestions. Your detailed comments have greatly helped us refine our work and clarify its contributions. In this response, we aim to address the common concerns comprehensively, highlighting the steps we have taken to improve the text.

**Contribution.** Our primary contribution lies in the development of a novel loss function specifically designed to address limitations in previous approaches, particularly in aligning predictions with ground truth events. We identify and empirically demonstrate that prior methods often exhibit misalignment in matching predictions to ground truth. DeTPP overcomes this issue by adapting the "set prediction" loss from object detection, incorporating several critical modifications necessary for the domain of event sequences. Finally, DeTPP achieves state-of-the-art (SOTA) results in long-horizon prediction, outperforming existing methods by a substantial margin.

**Novelty.** The novelty of our work spans three key aspects: problem formulation (Section 3), proposed solution (Section 4), and experimental evaluation (Section 5). First, we identify the alignment issues inherent in autoregressive and Next-K methods, providing an original analysis of these limitations. Second, we draw a novel analogy between object detection and event sequence forecasting, noting that, to the best of our knowledge, object detection techniques have not been previously applied to domains such as time series or recommendation systems. Furthermore, we introduce a probabilistic model for structured data, expressing the objective through likelihood maximization, a theoretical advance over prior approaches like DeTR [1]. By unifying matching and optimization objectives, we offer a coherent framework that is unique in this domain. We also highlight the necessity of calibration, which is essential for accurate forecasting. Lastly, we present a novel study on the relationship between prediction diversity and temperature values, offering a unique contribution to the field of event sequences.

**Experiments and baselines.** Our experimental evaluation includes a wide range of popular baselines that represent diverse methodologies. These include RNNs and Transformers, intensity-based and intensity-free approaches, next-event and horizon prediction methods, discrete and continuous-time models, including ODE-based approaches. Unfortunately, many recent works face challenges with formulation, reproducibility, or scale. For instance, the implementation of ContiFormer [2] is limited to a toy example—a spiral dataset—which requires over 2 hours of training on an RTX 4060 GPU, questioning the actual effectiveness of the approach. Furthermore, the authors have not addressed or commented on reproducibility concerns raised on GitHub. In contrast, we provide implementation, hyperparameters, and full evaluation results for ALL methods considered, achieving exceptional reproducibility in the event sequence domain.

**Domain importance and motivation.** The field of event sequence modeling remains underexplored, primarily due to the inherent sensitivity of the data involved. Event sequences are prevalent in domains such as finance (e.g., banking), retail, and healthcare, where data privacy concerns necessitate extensive anonymization, limiting its availability to the research community. Despite these challenges, modeling event sequences is a critical task for these industries. Accurate event modeling supports essential applications such as strategic planning, personalized communication, and advanced analytics. These capabilities yield significant financial benefits in sectors like finance and retail or, as for the MIMIC medical dataset, contribute to improved diagnostic accuracy. Our method achieves significant improvements over existing approaches in long-horizon prediction tasks, consistently outperforming prior methods by a wide margin. Furthermore, DeTPP+ demonstrates results comparable to or surpassing state-of-the-art (SOTA) methods in next-event prediction tasks. These findings highlight the significance of DeTPP across a wide range of applications.

[1] Carion N. et al. “End-to-end object detection with transformers”, ECCV, 2020

[2] Chen Y. et al. “Contiformer: Continuous-time transformer for irregular time series modeling”, NeurIPS, 2024

---

### Author Response · Authors · 2024-11-24
**Discussion**

Dear Reviewers,

We sincerely appreciate the time and effort you have devoted to providing detailed reviews and constructive suggestions. As the rebuttal phase comes to a close, we would like to check if our responses have addressed your concerns. If so, we would be grateful if you could consider updating your review and score. Additionally, if there are any remaining questions or points to discuss, please feel free to reach out.

Thank you again for your thoughtful feedback and valuable insights. We look forward to hearing your thoughts!

---

### Meta-Review · Area_Chair_AWNT · 2024-12-20

**Metareview:**

This work aims to address the issues of wrong matching and repetitive outputs in long-horizon events forecasting. However, the reviewers pointed out that there are a series of weaknesses. (1) Lack of novelty. The work adopted the Hungarian matching from DETR with minimal modifications and improvements. (2) Lack of comparison with some advanced approaches. (3) The paper is difficult to follow for readers without much background in this problem. Due to the shortcomings, all reviewers recommend rejection.

**Additional Comments On Reviewer Discussion:**

Reviewers pointed out that there are a series of weaknesses. (1) Lack of novelty. The work adopted the Hungarian matching from DETR with minimal modifications and improvements. (2) Lack of comparison with some advanced approaches. (3) The paper is difficult to follow for readers without much background in this problem. After rebuttal, reviewers are still unhappy about this work.

---

### Decision · Program_Chairs · 2025-01-22

Reject